# Advances in Liposomal Interleukin and Liposomal Interleukin Gene Therapy for Cancer: A Comprehensive Review of Preclinical Studies

**DOI:** 10.3390/pharmaceutics17030383

**Published:** 2025-03-18

**Authors:** Eman A. Kubbara, Ahmed Bolad, Husam Malibary

**Affiliations:** 1Clinical Biochemistry Department, Faculty of Medicine, Rabigh Branch, King Abdulaziz University, Rabigh 21911, Saudi Arabia; 2Department of Biochemistry and Molecular Biology, Faculty of Medicine, Al-Neelain University, Khartoum 11121, Sudan; 3Department of Microbiology and Unit of Immunology, Faculty of Medicine, Al-Neelain University, Khartoum 11121, Sudan; 4Department of Medicine, Faculty of Medicine, King Abdulaziz University, Rabigh 21911, Saudi Arabia

**Keywords:** liposomal interleukin, gene therapy, cancer immunotherapy, cytokine delivery, preclinical studies

## Abstract

Background: Preclinical studies on liposomal interleukin (IL) therapy demonstrate considerable promise in cancer treatment. This review explores the achievements, challenges, and future potential of liposomal IL encapsulation, focusing on preclinical studies. Methods: A structured search was conducted using the PubMed and Web of Science databases with the following search terms and Boolean operators: (“liposomal interleukin” OR “liposome-encapsulated interleukin”) AND (“gene therapy” OR “gene delivery”) AND (“cancer” OR “tumor” OR “oncology”) AND (“pre-clinical studies” OR “animal models” OR “in vitro studies”. Results: Liposomal IL-2 formulations are notable for enhancing delivery and retention at tumor sites. Recombinant human interleukin (rhIL-2) adsorbed onto small liposomes (35–50 nm) substantially reduces metastases in murine models. Hepatic metastasis models demonstrate superior efficacy of liposomal IL-2 over free IL-2 by enhancing immune responses, particularly in the liver. Localized delivery strategies, including nebulized liposomal IL-2 in canine pulmonary metastases and intrathoracic administration in murine sarcoma models, reduce systemic toxicity while promoting immune activation and tumor regression. Liposomal IL gene therapy, delivering cytokine genes directly to tumor sites, represents a notable advancement. Combining IL-2 gene therapy with other cytokines, including IL-6 or double-stranded RNA adjuvants, synergistically enhances macrophage and T-cell activation. Liposomal IL-4, IL-6, and IL-21 therapies show potential across various tumor types. Pairing liposomal IL-2 with chemotherapy or immune agents improves remission and survival. Innovative strategies, including PEGylation and ligand-targeted systems, optimize delivery, release, and therapeutic outcomes. Conclusions: Utilizing immune-stimulatory ILs through advanced liposomal delivery and gene therapy establishes a strong foundation for advancing cancer immunotherapy.

## 1. Introduction

### The Role of Interleukins (ILs) in Advancing Cancer Treatment

Cancer persists as a major global health challenge, posing a substantial barrier to improving worldwide life expectancy [1]. In the United States, estimates for 2024 indicate 2,001,140 new cancer cases and 611,720 cancer-related deaths [2]. Although traditional treatments, including chemotherapy, radiotherapy, and surgical interventions, have made advancements in cancer treatment, their lack of specificity frequently leads to severe and debilitating side effects, including hair loss, anemia, and immune suppression [3,4]. Chemotherapy, in particular, has limitations such as low water solubility, lack of targeted action, and the appearance of multi-drug resistance [5], leading to significant toxicity for both cancerous and healthy cells. Similarly, the untargeted nature of radiotherapy damages normal tissues [6].

To address these challenges, recent developments in cancer therapy have moved towards more precise and less harmful treatments, such as cancer immunotherapies [7,8,9]. A key component of these advancements is the use of cytokines, which serve as important factors in regulating the immune response. Cytokine-based treatments have proved crucial in activating the immune cells against tumors and remain a fundamental part of the ongoing cancer research [10,11]. Cytokines, including ILs, facilitate the immune cells to destroy tumors, with many of these molecules currently undergoing evaluation in clinical trials [12].

Cytokines are proteins produced in reaction to pathogen exposure, which is pivotal in mediating immune responses. ILs, a subset of cytokines, are produced transiently due to unstable messenger RNA and are rapidly secreted after synthesis. These molecules regulate cellular responses through up- and down-regulatory mechanisms. ILs also exhibit diverse roles; for example, IL-4, IL-5, and IL-13 act as essential factors for B cell growth and proliferation. Additionally, they aid in helper T cells’ differentiation into Th-1 and Th-2 groups and activate phagocytes. ILs frequently influence the production and action of other ILs; for instance, IL-1 activates lymphocytes and triggers IL-2 release. Most cytokines either act on the cell that produces them or on nearby cells. However, certain cytokines, such as IL-1, possess the capability to spread to distant locations and induce fever by affecting the central nervous system. Notably, small amounts of ILs are sufficient to elicit significant biological effects [13].

IL-2 was the pioneering cytokine granted authorization for the management of metastatic renal cell carcinoma and advanced melanoma [14,15]. Early clinical studies with elevated dosages of IL-2 demonstrated encouraging outcomes, with some patients showing tumor regression, representing a major step forward in immunotherapy [16,17]. However, despite these promising outcomes, systemic IL therapies such as IL-2 have been restricted by high toxicity levels and inconsistent results. Elevated doses of cytokines may result in significant adverse effects [18,19], including multiorgan dysfunction [20] and cytokine release syndrome, triggered by an overproduction of cytokines and increased inflammatory response due to IL-2 immune activation, with fatal side effects occurring in 2% of patients [21]. Similar challenges have been observed with IL-12, as its effective anti-tumor activity has been limited by its toxicity in clinical applications [22,23].

In response to these challenges, researchers have explored alternative delivery strategies to improve cytokine safety and efficacy. One promising approach is liposomal delivery, which uses lipid bilayer vesicles to encapsulate ILs, protecting them from degradation, enhancing stability, and targeting their delivery to tumors. This strategy reduces systemic exposure and associated toxicity while concentrating the therapeutic effects within the tumor microenvironment [24]. By leveraging liposomal technology, ILs can overcome the limitations of systemic administration, offering a more efficacious and safer option for cancer immunotherapy.

This review thoroughly examines the ongoing advancements in the liposomal encapsulation of ILs, focusing on preclinical studies, and explores the achievements, challenges, and future potential of this method in cancer therapy.

## 2. Clinical Potential of Liposomal Interleukin Formulations

Emerging clinical evidence highlights the promise of liposomal IL formulations in cancer therapy. A preliminary trial involving individuals with advanced melanoma showed that combining a liposomal melanoma vaccine with liposomal IL-2 produced notable outcomes. Specifically, the group receiving both the vaccine and low-dose localized liposomal IL-2 exhibited a 60% clinical response rate, with three patients achieving full remission and another three showing partial remission. In comparison, administering low-dose liposomal IL-2 alone resulted in partial responses for 60% of participants. Conversely, no clinical improvements were found in patients treated solely with the vaccine or with the vaccine paired with systemic IL-2. Furthermore, the combination approach triggered delayed hypersensitivity reactions, which were associated with enhanced activation of lymphocytes and tumor cell destruction, highlighting the synergistic benefits of liposomal IL-2 in cancer immunotherapy [25].

Further supporting the therapeutic potential of liposomal IL formulations, a phase I/II clinical trial was conducted to analyze the safety, optimal dosage, and anticancer potential of a Semliki Forest virus vector designed to deliver the IL-12 gene within CLs for treating recurrent glioblastoma multiforme. Preclinical research in breast and prostate cancer models verified the safety and anticancer effectiveness of this strategy. Moreover, early clinical trials in melanoma and renal cell carcinoma further confirmed its safety profile. These insights laid the foundation for its application in recurrent glioblastoma, underlining its potential for translation into clinical practice [26].

The practicality and safety of delivering liposomal IL via inhalation were studied in a phase I trial including patients having lung metastases. Using nebulized liposomal IL-2 administered three times per day, the study showed that doses as high as 6.0 × 10^6^ IU were well-tolerated without notable toxicity. This approach successfully stimulated localized immune responses in the lungs while reducing systemic effects, paving the way for future research on its therapeutic impact in metastatic lung cancer [27].

## 3. Overview of Liposomal Nanoformulations

Liposomal nanoformulations, with dimensions varying between 50 and 300 nm, are extensively utilized in biomedical disciplines due to their versatility and effectiveness in delivering therapeutic compounds [28,29,30]. These vesicles showcase structural variability, with sizes spanning from 0.025 μm to 2.5 μm, and can be composed of either a single or multiple layers of bilayer membranes. Liposomal size is crucial in influencing their circulation duration in the bloodstream, while both size and bilayer configuration influence their drug encapsulation capacity. Based on their structural characteristics, liposomes are classified as multilamellar vesicles (MLVs) or unilamellar vesicles (ULVs), the latter being further subdivided into large ULVs and small ULVs [31].

Progress in liposomal drug delivery technologies has resulted in four primary categories: conventional liposomes, ligand-targeted liposomes, sterically stabilized liposomes, and hybrid systems that integrate multiple strategies. Conventional liposomes, often referred to as the pioneering generation, consist of lipid bilayers composed of cholesterol and various lipids—cationic, anionic, or neutral—encapsulating an aqueous core [32].

### 3.1. Methods of Liposome Preparation

Liposomal preparation involves a series of standardized processes, including using organic solvents to dissolve lipids, eliminating the solvent through evaporation or drying, rehydrating the lipid film with an aqueous solution, and adjusting the particle size and bilayer structure as needed. Additional steps, such as purification, sterilization, and characterization, are essential to ensure the final product’s quality and functionality. Depending on the specific preparation technique, hydration may occur either prior to or following solvent removal [33].

A key aspect of liposome formation is the introduction of lipid molecules into an aqueous environment, irrespective of the desired size or structure. Studies conducted by Lasic provided a comprehensive description of lipid hydration methods and the resulting liposome structures [34,35].

### 3.2. Thin-Film Hydration Method

The thin-film hydration technique, often recognized as the Bangham method, is among the commonly employed approaches for producing MLVs. This process involves creating a thin lipid layer on the interior surface of a rotary evaporator flask, which is then hydrated using water or a buffer solution. By preheating both the lipid layer and the hydration medium to a temperature surpassing the lipid’s phase transition point, smoother bilayer development is facilitated. To detach the lipid film and generate MLVs of various sizes, vigorous stirring and sonication are applied [36].

Refinements suggested by Lasch, Weissig, and Lasic involve dissolving specific lipids, such as 1,2-Dipalmitoyl-sn-glycero-3-phosphocholine or 1-Palmitoyl-2-oleoyl-sn-glycero-3-phosphocholine, in a mixture of methanol and chloroform. The solvent is subsequently eliminated using a rotary vacuum evaporator, leaving behind a lipid film that is subsequently dried under high vacuum conditions until a stable weight is achieved. Hydration performed at temperatures surpassing the lipid’s phase transition point results in liposome suspensions with concentrations from 0.5 to 10 mg/mL. Processes such as intermittent shaking, sonication, and freeze–thaw cycles are utilized to enhance bilayer uniformity and stability [35,36,37].

### 3.3. Alternative Methods for Liposome Preparation

Innovative techniques, including freeze–thaw cycles, dehydration–rehydration, and reverse-phase evaporation, have been created to improve the encapsulation of proteins in liposomes. These methods improve the interaction of lipid bilayers with proteins, thereby increasing the effectiveness of encapsulation. For example, during the preparation of liposomal IL-2, synthetic 1,2-dimyristoyl-sn-glycero-3-phosphocholine is hydrated using phosphate-buffered saline that contains both ovalbumin and IL-2. The encapsulation process involves three rounds of freeze–thawing followed by sonication, resulting in encapsulation efficiencies ranging from 45% to 60%. Ensuring the freshness of the liposomes is essential for preserving their functional properties, especially for immunization purposes [38].

### 3.4. Role of Excipients in Enhancing Liposomal Stability

Liposomes are inherently unstable due to the fragile nature of their phospholipid membranes, which poses significant challenges for their industrial production. Enhancing their stability is essential to guarantee the sustained release of encapsulated active compounds, whether administered via injection or orally. The stability of liposomes is affected by physical variables, such as temperature and light during storage, as well as chemical factors like lipid oxidation and pH variations [39].

To deal with these concerns, several approaches have been employed. These include the application of protective coatings, structural modifications to the liposomal membrane, incorporation of cryoprotectants prior to lyophilization, and the application of surfactants or specialized polymer gels [40,41,42,43].

Cryoprotectants such as dimethyl sulfoxide, glycerol, sugars, disaccharides, and polyampholytes are particularly effective in protecting liposomes during storage under low-temperature conditions. By avoiding the development of ice crystals, these agents help preserve the structural integrity of liposomes [44].

Polymers such as chitosan, pectin, and alginate are frequently utilized as coatings to enhance liposomal functionality. These coatings offer several benefits, including the following: (1) prolonging the circulation time, thereby improving the transport and release of active compounds; (2) providing a protective barrier against oxygen, which minimizes lipid oxidation; and (3) reducing the leakage of encapsulated substances [45]. Furthermore, saccharides like chitosan, alginate, pectin, and starch have been shown to improve liposomal physicochemical and structural properties. Coated liposomes demonstrate enhanced biochemical stability under biological stressors, including pH fluctuations, temperature changes and osmotic pressure. These advancements result in prolonged release profiles, increased bioavailability, and improved biostability compared to uncoated liposomes [46,47,48].

Recent advancements in liposome technology have focused on modifying membranes with film-forming polymers like polyethylene glycol (PEG), Eudragit EPO, and poly(L-lysine). Among these, PEG is the most widely used, particularly in liposomes designed for antitumor drug delivery. Its hydrophilic nature, molecular flexibility, and neutral charge contribute to prolonged circulation times and reduced clearance by the reticuloendothelial system (RES) [49]. PEGylation, which involves the covalent attachment of PEG to liposomal surfaces, enhances steric hindrance. This prevents interactions with blood proteins, reducing opsonization and macrophage-mediated clearance and ultimately extending systemic circulation times [50,51].

The introduction of sterically stabilized liposomes, achieved by incorporating hydrophilic polymers like PEG, has significantly advanced liposomal drug delivery systems. These steric barriers mitigate in vivo opsonization by serum proteins, thereby reducing rapid recognition and uptake by the RES. This leads to prolonged circulation times, an enhanced accumulation of drugs at target sites, and minimized side effects [52,53,54].

Pharmacokinetic studies reveal that the half-life of sterically stabilized liposomes varies, ranging from 2 to 24 h in rodents and up to 45 h in humans, depending on particle size and polymer properties [32].

Initial approaches to enhancing liposome stability mimicked erythrocyte membranes by modifying liposomal surfaces with gangliosides, sialic acid derivatives (e.g., monosialoganglioside GM1), phosphatidylinositol, and neutral phospholipids. These modifications aimed to reduce interactions with blood components and improve stability [55]. The extensively studied steric stabilization mechanism reveals that PEG or GM1 molecules create a physical barrier near the liposome surface, reducing plasma protein binding, opsonization, and RES capture [56,57,58,59].

Despite its benefits, PEG has limitations. Low-molecular-weight PEG can lead to toxicity, and PEG coatings may become unstable when exposed to external stresses like heat, radiation, or mechanical forces, compromising liposomal stability and storage conditions [60,61]. Natural macromolecules, such as biodegradable proteins, offer a promising alternative to synthetic PEG. These coatings, including albumin, gelatin, and silk fibroin, enhance the prolonged storage of active compounds, promote continuous drug release, and improve targeting precision [62,63,64,65].

Combining different coating materials, such as chitosan–pectin, pectin–whey protein, and PEG–chitosan, has led to notable enhancement in liposomal physical and chemical characteristics. These combinations enhance solubility, thermal stability, oxidative resistance, and cellular protection [66,67,68,69]. Coatings made from dimethyl amino methyl-dextran, silica, and ceramics have demonstrated additional benefits, such as improved skin permeation, oxygen protection, photovoltaic stability, and anti-interference capabilities [70,71,72].

Helper lipids play a pivotal function in improving the performance of liposomes. Fusogenic lipids, such as Dioleoylphosphatidylethanolamine, facilitate the release of nucleic acids from endosomes by disrupting cellular membranes. Replacing this lipid with Dioleoylphosphatidylcholine, however, significantly reduces the efficiency of transfection, highlighting the importance of choosing the right lipid during formulation [73,74].

Cholesterol also plays a pivotal role in enhancing liposomal stability. By promoting a more compact arrangement of phospholipids, cholesterol improves the mechanical properties of liposomes, increases drug entrapment efficiency, and reduces lipid area per molecule, as demonstrated by studies showing significant improvements in liposome compressibility [75,76,77,78,79]. Moreover, cationic cholesterol derivatives like 3β-[N-(N’,N’-dimethylaminoethane)-carbamoyl] cholesterol have been investigated as substitutes for cationic phospholipids, showing improved efficacy in delivering nucleic acids in various systems [80,81].

### 3.5. Cationic Liposomes for Gene Delivery

CLs include lipids carrying positive charges like 1,2-Dioleoyl-3-trimethylammonium-propane and are commonly utilized for delivering nucleic acids. These liposomes create electrostatic complexes, known as lipoplexes, by interacting with nucleic acids that carry negative charges. This interaction safeguards nucleic acids from enzymatic breakdown and enhances cellular uptake via electrostatic interaction with negatively charged membranes. Once inside the cell, the nucleic acids are discharged into the cytoplasm following endosomal escape, facilitated by the destabilizing action of cationic lipids on the endosomal membrane [82,83,84].

It was noted that the freeze–thaw encapsulation was the most effective procedure regarding liposomal encapsulation of nucleic acids [85]. Cationic lipids, possessing fixed positive charges, enhance ionic interactions with negatively charged cell membranes, but due to their fixed positive charge and amphiphilic nature, they exhibit cytotoxicity, which disrupts the lipid bilayer of cell membranes. To address this issue, neutral amino lipids were designed to minimize this cytotoxic effect [86].

### 3.6. Toxicity Studies of Liposomal Formulations

Liposomes are primarily composed of phospholipids, cholesterol, and PEG, with each component contributing uniquely to their functionality. Phospholipids, the main structural elements, encompass various molecular types and isomers. Commonly utilized phospholipids in liposome production include lecithins derived from eggs and soybeans, along with sphingomyelins [87].

Upon entering the bloodstream, plasma proteins adhere to the surface of liposomes. Research has shown that surface properties such as charge and PEGylation significantly influence the adsorption of blood proteins, forming a “protein corona.” This protein layer performs a vital role in mediating interactions between liposomes and both healthy and diseased cells [88,89,90].

A multitude of research efforts have investigated the adverse effects and toxicity associated with liposomes. In 1973, Papahadjopoulos et al. [91] investigated the impact of liposomes on the cytotoxicity of healthy L929 mouse fibroblast cells. They found that incorporating 3% or 10% lysophosphatidylcholine (lysolecithin, lysoPC) into liposomes enhanced intracellular uptake, increasing polykaryocytosis by approximately 7%. This modification notably reduced cell viability, with survival rates dropping from 97% to less than 37% following liposomal incubation.

Adams and colleagues conducted one of the earliest in vivo studies on liposomal toxicity in 1977 [92].

Although the primary aim of liposomal drug delivery is to minimize chemotherapeutic toxicity and enhance targeted delivery, liposomes and their components may induce toxic or allergic reactions and immune responses. While PEG conjugates provide significant benefits, studies have revealed unexpected immune reactions to PEGylated nanocarriers, including accelerated blood clearance (ABC) and complement activation-related pseudoallergy (CARPA). The ABC effect is triggered by the production of anti-PEG antibodies following the first injection, resulting in rapid elimination during subsequent administrations. CARPA, a hypersensitivity reaction, compromises the safety and efficacy of PEGylated pharmaceuticals [93,94,95,96,97].

Despite these concerns, free PEG is widely recognized as safe, earning Generally Recognized as Safe (GRAS) status from the U.S. FDA. However, the immunogenicity of PEG remains debated. Studies suggest that anti-PEG antibodies primarily target antigenic determinants at the interface between PEG and other materials rather than PEG itself [98].

Cationic liposomes, commonly used for delivering nucleic acids, have been reported to exhibit toxicity in immune cells, including macrophages [99]. Takano et al. and Aramaki et al. [100,101] demonstrated that the cellular uptake of cationic liposomes by the mouse macrophage-like cell line RAW264.7 is associated with apoptosis and the production of reactive oxygen species.

Further insights into liposomal toxicity were provided by Kedmi et al. [102], who investigated hepatotoxicity and inflammatory responses using in vivo animal models. Their study revealed significant weight loss, elevated hepatic enzyme levels, and a marked increase in Th1- and Th17-dependent cytokines within two hours of liposome injection in C57BL/6 mice. Similarly, in a rat model, DOTAP/CHOL liposomes were found to upregulate oxidative stress-related genes such as HMOX1 and OGG1, leading to DNA strand breaks in the lung and liver [103].

Although cationic lipids alone exhibited toxicity, the addition of DOPE amplified these effects. However, substituting DOPE with dipalmitoylphosphatidylcholine significantly reduced macrophage toxicity. Moreover, incorporating dipalmitoylphosphatidylethanolamine–PEG2000 (DPPE-PEG2000, 10 mol%) completely eliminated this toxicity [104]. Moreover, mitigation strategies for the adverse effects of cationic liposomes were also demonstrated by Soenen et al. [105], who used reactive oxygen species scavengers or calcium channel blockers in 3T3 fibroblasts.

### 3.7. Liposomal Interleukin Formulations

Encapsulating proteins such as IL-2 within liposomes presents challenges due to their large molecular size and complex nature. Encapsulation methods are divided into two types: passive loading, where proteins are incorporated during the liposome formation process, and active loading, which involves the use of preformed liposomes to encapsulate proteins via a remote-loading technique [106].

For IL-2 encapsulation, a thin lipid film is hydrated with an IL-2-containing solution to produce MLVs. The efficiency of encapsulation is strongly affected by determinants such as liposome surface charge, composition, pH, and ionic conditions. Negatively charged liposomes, formulated with specific phosphatidylcholine-to-phosphatidylglycerol ratios, demonstrate improved encapsulation efficiency in acidic environments and when stabilizers like glycerol and bovine serum albumin are present. The combination of IL-2-loaded liposomes with immunostimulatory agents enhances immune responses, offering promising potential for cancer immunotherapy [107].

## 4. Overview of Liposomal Drug Delivery Systems

Originally discovered by Alec Bangham in 1963, liposomes are some of the most widely researched nanostructures for innovative drug delivery solutions [108]. These vesicles, comprising concentric lipid bilayers enclosing aqueous cores, can encapsulate both hydrophilic and hydrophobic compounds, making them flexible vehicles for transporting variety of drugs. Hydrophobic compounds and molecules are enclosed within the bilayer membrane, whereas hydrophilic ones are confined to internal aqueous compartments [32,109]. This characteristic of liposomes offers several advantages, such as accurate targeting, regulated release, enhanced stability, and lower toxicity, compared to traditional methods of drug administration [110].

Paul Ehrlich’s “magic bullet” concept, introduced in the late 19th century, laid the groundwork for selective drug delivery methods. Ehrlich envisioned chemical carriers that could specifically target malignant cells while sparing healthy tissue. Expanding on this idea, liposomes have been developed into a noteworthy tool for delivering medications and drugs, especially in cancer therapy. Encapsulating anticancer medications within liposomal structures provides a safe and advanced approach for targeted treatments, reducing cytotoxic effects on healthy cells and ensuring that higher drug concentrations reach tumor sites [111,112]. However, the effective delivery of nanoparticles into solid tumors continues to pose a significant challenge, reducing the effectiveness of many therapies [113].

Liposomal drugs such as Doxil (doxorubicin), Lipusu (paclitaxel), and Marqibo (vincristine) have already been integrated into clinical practice for cancer therapy [114]. Recent regulatory approvals of liposome-formulated drugs demonstrate their ability to improve therapeutic outcomes by targeting specific tissues and minimizing exposure to sensitive areas [115]. Some formulations, such as Doxil, have been improved by adding targeting ligands such as monoclonal antibodies (e.g., mAb 2C5) or employing strategies such as hyperthermia in ThermoDox to trigger drug release [116].

Notably, liposome stability, clearance, and biodistribution in the bloodstream rely on many factors, including size, composition, and charge. Smaller, rigid liposomes (100–200 nm) are more likely to remain intact in circulation without degradation [117]. Liposomal size has also been found to influence uptake and biodistribution in the lymphatic system following subcutaneous injection. Smaller liposomes (~0.04 µm) demonstrate higher uptake when contrasted with larger, non-localized liposomes, which tend to remain near the injection site. Liposomal composition is also significant, with phosphatidylserine-containing liposomes showing improved localization in lymph nodes [118]. To improve circulation time and decrease macrophage uptake, PEG has been added to the liposomes using PEGylation. PEGylated liposomal IL-2 formulations, for example, have shown reduced accumulation in the kidneys and liver while increasing localization in tumor tissues [119]. However, PEGylation and lipids may reduce the biological efficacy of liposomal IL-2 by concealing the cytokine and hindering its ability to attach to membrane-bound receptors [120,121]. Nonetheless, the liposomal encapsulation of IL-2 notably prolongs its serum half-life, enhancing its retention in the rough endoplasmic reticulum and lymphatic systems and potentially enhancing its immune-stimulating effects [122].

The enhanced permeability and retention (EPR) effect, where nanoparticles around 100 nm in size permeate across the blood vessels and concentrate in cancerous tissues, is an important factor in the effectiveness of liposome-based delivery [123,124]. For instance, large ULVs, produced by extruding MLVs through 100 nm polycarbonate membrane filters [125], have been utilized to increase drug delivery accuracy. In some cases, oligolamellar vesicles, which consist of two or three bilayers, have been produced using techniques such as reverse-phase evaporation [126] or ethanol-based proliposome technology [127], facilitating greater flexibility in drug delivery.

A preclinical study using xenograft models of head and neck squamous-cell carcinoma in immunocompromised rats indicated that intertumoral administration of liposomal IL-2 significantly enhanced drug retention and diffusion within the tumor. Planar gamma camera imaging showed that 39.2% of 99mTc-labeled liposomes stayed in the tumor after 20 h, in contrast to only 18.7% for non-liposomal 99mTc-complex [128]. This outcome emphasizes the superior performance of liposomal formulations in achieving greater tumor retention and spread, which is crucial for achieving effective localized therapy in solid tumors.

Although liposomal delivery provides various advantages, including targeted drug delivery and lower toxicity, it presents certain challenges, especially regarding immune responses. When liposomes are administered systemically, they can be eliminated by mononuclear phagocytes. In certain instances, macrophages accumulate lipid droplets, which activate inflammatory pathways such as the nuclear factor kappa-light-chain-enhancer of activated B cells signaling cascade, stimulating cytokine production of IL-1β and IL-6. This inflammatory reaction may influence the normal function of macrophages in the bone marrow, possibly resulting in biased myeloid hematopoiesis and reduced red blood cell production [129].

Despite these challenges, liposomes have proven invaluable in cancer treatment and various pharmaceutical, cosmetic, and biochemical applications. Their high compatibility makes them ideal carriers for delivering drugs, peptides, proteins, plasmid DNA, and oligonucleotides [130]. Ongoing advancements, including liposomes specifically designed to target cancer cells, have notable potential for enhancing therapeutic efficacy and reducing side effects [131,132]. 

By utilizing the distinct characteristics of liposomes and modifying their surface properties, researchers are improving their effectiveness in drug delivery systems. As mentioned earlier, PEGylated liposomes can escape immune detection, and formulations with targeting antibodies have increased the therapeutic potential of these nanocarriers. Early versions of liposomes were rapidly removed by the reticuloendothelial system, but advancements have resulted in the development of more stable long-circulating liposomes that build up at the intended target locations through the EPR effect [133]. Furthermore, the application of nanoparticles, both naturally occurring (e.g., extracellular vesicles and lipid-based nanoparticles) and synthetic (e.g., polymeric, metallic, and silica-based), is advancing, presenting exciting opportunities for clinical diagnostics and treatments [134].

Moreover, the advancement of enhanced liposomal delivery systems signifies the integration of nanotechnology with medicine, with the primary purpose of optimizing human health outcomes and improving quality of life [135].

## 5. Liposomal Interleukin Therapy

The liposomal delivery of IL-2 has gained recognition for its potential to increase the effectiveness of IL-2-based therapies while reducing related toxicities. Liposomes have been demonstrated to enhance the growth of cytotoxic T-lymphocyte line cells, which are commonly utilized to assess IL-2 activity. This effect is more noticeable at low IL-2 doses and reduced lipid concentrations [136]. Moreover, the liposomal encapsulation of IL-2 has also been demonstrated to improve its therapeutic effectiveness in treating diseases such as cancer and acquired immunodeficiency syndrome by extending its half-life and minimizing related side effects [124]. For example, aerosolized liposomal IL-2 formulations have been evaluated for their potential in treating pulmonary metastases and primary lung cancers, showing stability during nebulization and distribution throughout the lungs [137].

Additionally, in various experimental models, rhIL encapsulated in liposomes has shown enhanced tumor growth inhibition and improved antitumor efficacy, further emphasizing the importance of proper formulation characterization for clinical applications [138]. For instance, in a rat hepatoma model, liposomal IL-2 significantly increased survival time compared to free IL-2 or saline, largely due to enhanced macrophage activation and tumor necrosis factor alpha (TNF-α) production [139]. In a previous study, the combination of glycosylated rhIL with liposomal bilayers has shown that negatively charged liposomes, created under specific pH conditions, achieved the highest incorporation efficiency [109].

Similarly, liposomal IL-2, when administered intravenously, showed extended presence in the bloodstream and increased levels in the liver and spleen in contrast to free IL-2. This effect was also observed following subcutaneous administration, where liposomal IL-2 demonstrated a longer residence time in the bloodstream than free IL-2 [140].

Research has demonstrated that liposome-based delivery of IL-2 greatly enhances its pharmacokinetics and biological activity in vivo. In one comparative study, rhIL-2 was enclosed into two different liposome-based formulations: large MLVs or conventional liposomes with a mean diameter of 0.75–1.5 μm and small, unilamellar, long-circulating liposomes (sterically stabilized liposomes [SSL]) with an average diameter of 60 nm. Both formulations surpassed free IL-2 in enhancing the growth of spleen cells and activating lymphokine-activated killer (LAK) cells, especially at low cytokine concentrations. Among these, MLV-IL-2 displayed better immunomodulatory properties, despite the longer circulation time of SSL-IL-2 [141]. Additionally, SSLs have shown potential as effective carriers for IL-2, providing high encapsulation efficiency, enhanced stability, and stronger immunomodulatory and anticancer outcomes. SSL-IL-2 has also been demonstrated to be significantly more stable than soluble IL-2, retaining its biological activity in vivo and highlighting its promise as a superior therapeutic agent [142]. 

Moreover, SSL-IL-2 needed reduced doses and less frequent administrations to achieve therapeutic outcomes, thereby reducing systemic toxicity compared to pegylated IL-2, which, although potent, was associated with higher toxicity [143]. Zhang et al. [144] showed that a therapeutic approach combining anti-CD137 and an IL-2-Fc fusion protein produced potent anticancer effects but at the same time contributed to severe immunotoxicity due to the excessive stimulation of circulatory white blood cells. To address this toxicity, the researchers attached this combination to a liposomal surface, enabling these immune activators to rapidly concentrate at tumor sites while reducing systemic toxicity. This immunoliposome delivery system produced comparable antitumor results to free agents in multiple tumor models without the associated systemic toxicity. It also facilitated the infiltration of tumors by cytotoxic lymphocytes, improved cytokine release, and increased the biosynthesis of granzyme, demonstrating similar immune-stimulatory impacts within the tumor microenvironment. Comparable outcomes were observed in another study, in which combining anti-CD137 with an engineered IL-2-Fc fusion protein attached to PEGylated liposomes demonstrated significant antitumor effectiveness in a B16F10 melanoma model. This method reduced systemic inflammatory toxicity and led to systemic immunity that prevented the growth of distant tumors [145].

Recent progress in cell-based delivery systems, such as Treg cell-mediated transport of drug-loaded liposomes to tumor cells, provided an additional promising strategy to enhance cancer immunotherapy, especially through the delivery of IL-2 and immune checkpoint inhibitors [146]. However, despite these advantages, liposome-encapsulated IL-2 (Lip-IL-2) is less effective than free and natural or recombinant IL-2 in stimulating LAK cells from normal donors. Nonetheless, Lip-IL-2 maintained LAK cytotoxicity in pre-activated cell populations, indicating that although Lip-IL-2 interacts similarly to free IL-2 with activated T cells, it is less efficient in activating LAK-cell precursors [147].

## 6. Preclinical Studies on Liposomal Interleukin Therapy

### 6.1. Liposomal IL-2 for Cancer Treatment in Preclinical Studies

Liposomal formulations of IL-2 have demonstrated significant promise for cancer therapy in preclinical studies, providing enhanced biodistribution, improved therapeutic efficacy, and reduced toxicity. One example includes rhIL-2 effectively adsorbed onto small hydrophobic liposomes (i.e., 1,2-Distearoyl-sn-glycero-3-phosphocholine (DSPC) and 1,2-Distearoyl-sn-glycero-3-phosphoglycerol (DSPG)., ranging from 35 to 50 nm in size) at a molar ratio of 10:1 DSPC: DSPG, with an IL-2-to-liposome ratio of 4.0 JRU/nmol lipid. This formulation significantly enhanced the dissemination of IL-2 following intravenous injection in mice and effectively reduced M5076 metastases [148].

Systemic delivery of liposomal IL-2 has also been examined in models of hepatic metastases. In a previous study, recombinant IL-2 adsorbed onto DSPC-DSPG liposomes showed better inhibition of hepatic metastases than free IL-2 in mice with M5076 tumors. The liposomal formulation outperformed free IL-2 at lower doses, indicating improved biodistribution and efficacy, particularly in the liver [149]. Furthermore, galactose-entrapped liposomes targeting liver metastases enhanced IL-2 accumulation in the liver and increased its antitumor activity while promoting cytotoxicity in hepatic sinusoidal lymphocytes [150].

Localized delivery strategies using liposomal IL-2 have also shown potential in reducing systemic toxicity. In dogs, nebulized liposomal IL-2 formulations, including IL-2-loaded liposomes, increased the activation of effector leukocytes in bronchoalveolar lavage, enhancing local immune responses without the adverse effects observed with free IL-2 [151]. In another study, intrathoracic administration of liposomal IL-2 has also shown superior antitumor effects. In mice with MCA-106 sarcoma lung metastases (C57BL/6 strain), administering IL-2 liposomes via intrathoracic delivery enhanced survival and demonstrated antitumor effects [152]. Similarly, in the B16 melanoma model, localized delivery of liposomal IL-2 into tumors promoted CD3+ T-cell infiltration and provided systemic defense against secondary tumors. Mice treated with liposomal IL-2 showed enhanced survival rates and immune memory, whereas free IL-2 failed to produce these effects [153].

The combination of anti-CD3-activated splenocytes and liposomal IL-2 was explored, particularly in post-bone marrow transplant settings. This approach enhanced CD8+ T-cell expansion and significantly improved survival in lymphoma-bearing mice, outperforming treatments with either IL-2 or splenocytes alone [154]. Liposomal IL-2 has also been combined with anti-CD3-activated T cells in murine models, especially when pretreated with cyclophosphamide. CD4+ T cells activated using anti-CD3, followed by liposomal IL-2 administration, achieved tumor remissions and extended disease-free survival in mice with MC-38 colon adenocarcinoma, 3LL lung carcinoma, or 38C13 lymphoma [155]. Similarly, in murine colon adenocarcinoma models, IL-2 liposomes combined with T-activated killer cells enhanced the infiltration of Lyt-2+ lymphocytes into hepatic metastases, resulting in potent antitumor effects [156].

In the Renca murine renal carcinoma model, daily localized delivery of IL-2 liposomes near the tumor site suppressed tumor progression and extended survival. This slow-release system increased spleen cell cytotoxicity and recruited tumor-infiltrating lymphocytes, including Lyt-2+ and L3T4+ cells, into the tumor, indicating that liposomal IL-2 stimulates a strong localized immune response [157].

Peritumoral delivery of IL-2 liposomes in rats with solid tumors induced by ascites-forming hepatoma (AH-66) cells led to substantial suppression of tumor growth, with immunohistochemical analysis revealing the accumulation of activated macrophages around the tumor site, contributing to enhanced antitumor responses [158].

Moreover, liposomal IL-2 combined with other therapeutic agents has proven effective. For example, pairing polyethylene glycol-coated liposomal doxorubicin (Doxil) with liposomal IL-2 for treating M109 lung adenocarcinoma in mice produced prolonged tumor-free survival in nearly all cases, markedly outperforming chemotherapy alone. Notably, liposomal IL-2 was most effective in intraperitoneal tumor models, with the combination therapy showing no significant toxicity, reinforcing the potential of liposomal chemoimmunotherapy for treating metastatic and regionally spread tumors [159]. Another combination therapy involved rhIL-2 with long-circulating, temperature-sensitive liposomes containing adriamycin (ALTSL) in mice bearing H22 tumors. This strategy significantly enhanced natural killer (NK) cell activity, promoted tumor cell apoptosis, and increased lymphocyte transformation rates while reducing adriamycin-induced toxicity [160].

Various types of different liposomal formulations, including cationic, anionic, and neutral liposomes, have been explored for delivering IL-2 to cervical cancer cells, with CLs demonstrating high affinity for cancer cells and enhanced IL-2 cytotoxicity [161]. Small hydrophobic liposomes carrying IL-2 showed enhanced distribution and therapeutic efficacy in treating metastases by extending the time IL-2 circulation time, which could lead to fewer injections and improved cancer treatment outcomes.

Moreover, PEGylated liposomes containing IL-2 were utilized in adoptive cell transfer (ACT), facilitating repeated targeting and stimulation of transferred T-cells in vivo. This technique notably enhanced T-cell proliferation while minimizing systemic toxicity, offering an innovative method to improve ACT effectiveness [162]. Similarly, oncoplin, which is a liposomal formulation of human serum albumin and IL-2, effectively increased surface IL-2 expression and targeted T and NK cells, facilitating the extended release of IL-2 and offering significant benefits over soluble IL-2 in immunotherapy applications [163].

### 6.2. Liposomal IL-1, IL-12, IL-13, and IL-15

Liposomal delivery systems have also shown promise beyond IL-2, extending to other ILs, such as IL-12. Encapsulating IL-12 in large, multilamellar liposomes was demonstrated to reactivate tumor-infiltrating T cells in human tumor xenograft models. Localized delivery of IL-12 caused significant growth of quiescent T cells and increased interferon (IFN)-γ production, leading to tumor cell apoptosis. Importantly, this method minimized systemic IL-12 levels, thereby reducing potential toxicity [164]. A recent research paper demonstrated that local administration of a combination of mRNA-carrying genes for IFN-α, IL-7, and IL-12 via nanoparticles reduced tumor size and long-lasting immune response against tumors. These effects were further amplified by combining it with immune checkpoint blockers [165]. However, IL-12 therapies using layer-by-layer nanoparticles for treating ovarian cancer demonstrated reduced toxicity and more antitumor effectiveness than layer-free liposomes [166].

Beyond IL-12, innovative combinations of ILs with other immune-stimulating agents have been explored. For instance, combining IL-12 with CLs and monophosphoryl lipid A in murine breast cancer models enhanced cytolytic T-cell responses and macrophage activation. This combination suppressed tumor growth and initiated systemic immune responses, showcasing the synergistic effects of combining ILs with adjuvants in cancer immunotherapy [167].

Another example involves the co-administration of IL-15 and gemcitabine using folic acid-modified liposomes. In a mouse model, IL-15 encouraged CD8+ T and NK cell proliferation, whereas gemcitabine induced immunogenic cell death, enhancing NK cell recognition and CD8+ T cell activation. This combination effectively overcame immune evasion mechanisms and achieved significant antitumor effects [168].

The application of liposomal delivery systems has also extended to brain tumor therapy. IL-13-conjugated liposomes were developed for targeted delivery to glioblastoma multiforme, a highly aggressive brain tumor. This targeted approach improved drug accumulation and retention within glioma cells, leading to increased cytotoxicity and reduced tumor volume compared to non-targeted liposomal treatments [169].

Furthermore, a combination of TNF-α and IL-1α encapsulated in phosphatidylcholine and phosphatidylserine multilamellar liposomes showed effective targeting of cancer cells. This combination enhanced the survival of mice with metastatic melanoma and decreased metastatic lung tumor nodules. The encapsulated cytokines retained their biological functions, effectively promoting thymidine uptake in T-helper lymphocytes and enhancing target cell destruction [170]. These results emphasize the potential of liposomal cytokine delivery in decreasing tumor burden and enhancing survival rates in cancer models (Figure 1).

## 7. Liposomal Interleukin Gene Therapy for Cancer Treatment

Liposomal IL gene therapy has gained recognition as an effective method in cancer immunotherapy, harnessing cytokines to activate the immune system against tumors. Various liposome types have been utilized to deliver therapeutic genes, including neutral, cationic, polymer-coated, and ligand-targeted liposomes [171]. Among these, CLs are particularly effective due to their positive charge, acquired through amines in the polar head group. These positively charged amines bind strongly with anions, such as DNA, through both electrostatic and Van der Waals interactions, affecting the liposomes’ shape and stability [172].

CLs are spherical in structure and primarily consist of cationic lipids, which are effective at condensing nucleic acids, including mRNA, plasmid DNA (pDNA), and small interfering RNA (siRNA) [173]. CL-DNA complexes, which can interact with negative charges on cell membranes, are also being investigated for localized and systemic therapeutic gene transfer. These complexes have shown significant potential for delivering cytokine genes, including IL-2, IFN-α, and IL-12, directly within cancer cells, potentially inducing an antitumor immune response [174].

### 7.1. Liposomal IL-2 Gene Therapy

CLs composed of DOTMA liposomes and cholesterol were investigated to deliver hIL-2 genes to tumor cells. Enzyme-linked immunoassays conducted on tumor extracts 24 h after treatment revealed a significant rise in local hIL-2 production and the secondary activation of murine IFN-γ and IL-12, enhancing immune activity at the tumor location. In a murine model of head and neck tumors, the combination of partial surgical resection with non-viral hIL-2 gene therapy exhibited significant antitumor efficacy without any toxicity [175]. These results suggest that localized cytokine production following IL-2 gene transfer is crucial for therapeutic success.

IL-2 gene therapy was further explored using CLs (DMRIE/DOPE) in an orthotopic mouse model of bladder tumors. IL-2 gene therapy led to sustained tumor-free survival in 40% of the treated animals, with resistance to tumor recurrence, suggesting the development of immunological memory [176]. Moreover, intravesical liposomal IL-12 gene therapy outperformed Bacillus Calmette–Guerin therapy, providing long-term tumor-specific immunity and improved survival in bladder cancer models [177].

Similarly, a DNA plasmid vector carrying the IL-2 gene, pVCL-1102, was assessed both in vitro and in vivo. The development of plasmid with DMRIE/DOPE improved gene transfection efficiency and demonstrated notable antitumor activity in the B16 melanoma model despite challenges in distinguishing IL-2-specific effects from non-specific plasmid effects [178].

Pairing IL-2 with IL-6 has demonstrated enhanced immune responses. In a study exploring the impact of liposomal IL-2 and IL-6 gene therapy in mice with lymphoma, macrophage cytotoxicity and cytokine production significantly increased, indicating a combined effect of IL-2 and IL-6 in stimulating antitumor immune responses [179]. Moreover, the combined delivery of liposomal IL-2 and IL-6 genes significantly reduced cancer growth and extended survival, with enhanced synthesis of lymphocyte function associated with antigen 1 and major histocompatibility complex (MHC)-I on tumor-infiltrating lymphocytes, reinforcing the effectiveness of dual cytokine gene therapies [180]. Similarly, a study on EL-4 lymphoma-bearing mice showed that co-delivery of IL-2 and IL-6 DNA liposomes via intraperitoneal injection increased cytotoxicity, MHC-I expression, IL-1, and TNF-α levels. Notably, IL-2 treatment enhanced macrophage activation, and its combination with IL-6 produced the highest level of macrophage activation, indicating a synergistic antitumor effect [181].

Double-stranded RNA has also been explored as an adjuvant to IL-2 gene therapy due to its multiple antitumor mechanisms. Rodriguez et al. [182] used a plasmid encoding Sindbis viral RNA replicase (pSIN-IL2) combined with sigma receptor-targeted liposomes. This combination effectively suppressed the growth of B16 melanoma tumors and activated CD4+ and CD8+ T cells and NK cells, indicating its potential in melanoma therapy.

Moreover, BNT153 (an mRNA formulation encapsulated in liposomes), designed to encode the native form of IL-2, was tested in combination with BNT152 (an mRNA formulation of IL-7) in a phase 1 clinical trial; however, to date, no results from this trial have been published [183].

Another experiment using IL-2 gene therapy has demonstrated potential in treating head and neck squamous-cell carcinoma (SCCHN). In SCCHN cell lines transfected with IL-2 plasmids encapsulated in DOTMA/Col liposomes, bioactive IL-2 was produced for up to 30 days, leading to enhanced immune-stimulatory properties and release of ILs, including IL-6, IL-8, granulocyte macrophage colony-stimulating factor, and TNF-α. This study suggests that IL-2 gene therapy could effectively stimulate autologous immune responses in SCCHN, offering potency for clinical application [184].

### 7.2. Liposomal IL-4 and IL-12 Gene Therapy

Beyond IL-2, other cytokines, such as IL-4 and IL-12, have shown promise in gene therapy approaches. IL-4, with pleiotropic effects, was examined in a melanoma model using a combination of IL-4 and *Escherichia coli* cytosine deaminase suicide gene. This two-gene liposomal therapy substantially enhanced tumor regression and survival compared to single-gene therapy [185]. Additionally, IL-4 DNA combined with CLs, when injected into B16(F10) melanoma tumors, led to notable tumor regression through immune-mediated pathways [186]. Moreover, combining liposomes loaded with sorafenib and IL4R-targeting peptides complexed with microbubbles markedly inhibited the growth of A498 kidney cancer cells in an IL4R-dependent manner [187].

Furthermore, IL-12 and H-2Kb gene expression in malignant cells led to decreased cancer cell proliferation, enhanced infiltration of immune cells, and tumor necrosis production, showcasing the synergistic potential of these cytokines [188,189], Similarly, in metastatic liver tumor models, using all-trans-retinoic acid-cationic liposomes (ATRA- CLs) to deliver the IL-12 gene substantially decreased tumor size and improved survival outcomes. The addition of ATRA increased TNF receptor-1 expression and apoptosis in tumor cells, highlighting the potential of combining retinoic acid with cytokine gene therapies for enhanced therapeutic outcomes [190]. The same effect was also demonstrated in a B16 melanoma model, where the combination of paclitaxel and adenovirus encoding murine IL-12 (Ad5-mIL-12) delivered via anionic liposomes showed increased tumor suppression and extended survival. This combination therapy also raised serum and tumor IFN-γ levels, indicating a strong systemic immune response [191]. Similarly, in another study, IL-12 gene therapy delivered through CLs effectively suppressed tumor growth and extended survival, highlighting the ability of IL-12 to enhance antitumor immunity [192]. Additionally, IL-12 gene delivery using CLs in a C57BL/6 murine melanoma model substantially decreased tumor growth, enhanced survival, and enhanced NK cell activity [193].

A novel approach involving the freeze-drying of a monophase solution was employed to encapsulate murine IL-12 (mIL-12) pDNA into nanoliposomes. The formulated nanoliposomes had an average particle size of approximately 300 nm and a zeta potential of 96.5 mV. Their therapeutic efficacy was evaluated in vivo over 40 days in BALB/c mice bearing CT-26 colon carcinoma tumors through localized injections directly into the tumor. The nanoliposomes exhibited minimal cytotoxicity and achieved a remarkable 25-fold increase in mIL-12 expression compared to naked mIL-12 pDNA. Repeated administration of the mIL-12 pDNA-loaded nanoliposomes significantly reduced tumor growth, demonstrating their potential as an effective strategy for cancer gene therapy [194].

Non-viral gene delivery methods, such as sonoporation utilizing ultrasound-sensitive liposomes, are being explored to improve gene delivery efficiency. In combination with IL-12 gene therapy, these delivery systems dramatically suppressed tumor growth, with T-cell migration observed in treated mice, offering promise for future cancer gene therapy development [195].

### 7.3. Liposomal IL-15, IL-21, and IL-30 Gene Therapy

In a promising study, Caspy2 and IL-15 genes were co-expressed in murine tumors using CLs. This combination effectively reduced tumor growth, minimized spontaneous metastasis to the lungs, and triggered sustained immune protection against tumor recurrence. The enhanced apoptosis and immune response induced by this combination further highlight the potential of multi-agent gene therapies in cancer treatment [196]. IL-15 gene therapy has also demonstrated efficacy in orthotopic murine bladder cancer models, reducing tumor weight and promoting CD8+ T-cell infiltration, contributing to long-term resistance to tumor rechallenge [197].

A complementary study in a B16-F10 melanoma lung metastasis model using CLs to deliver an IL-15 plasmid showed significantly inhibited lung metastasis, enhanced splenic cell-mediated cytotoxicity, and increased NK cell recruitment, suggesting that IL-15 gene therapy holds promise for metastatic melanoma [198]. Furthermore, cationic protamine/liposome was designed to enhance mRNA delivery, offering efficient condensation and robust mRNA transport in vivo. The findings revealed that the cationic protamine/liposome system successfully facilitated the delivery of IL-15-encoding mRNA to C26 colorectal cancer cells, resulting in high transfection efficiency. The expressed IL-15 activated lymphocytes, enhancing immune-mediated cytotoxicity against tumors in vitro. Moreover, both localized and systemic administrations markedly suppressed tumor growth in various C26 murine colon cancer models [199].

In chronic myeloid leukemia, dendritic cell (DC)-mediated immunotherapy combined with cytokine gene therapy has shown promise. Researchers constructed a eukaryotic expression vector containing the genes for glucocorticoid-induced TNF receptor ligand (GITRL) and IL-21, which were introduced into DCs through liposomal delivery. These modified DCs significantly enhanced NK cell-mediated cytotoxicity and upregulated IL-2 and IFN-γ secretion, suggesting a novel therapeutic approach for patients with chronic myeloid leukemia who do not show a response to tyrosine kinase inhibitors [200] (Figure 2).

Advancing the therapeutic potential of liposomal interleukin-based systems, a recent study highlighted a novel immunoliposome platform for delivering CRISPR/Cas9 components targeting the IL30 gene, an IL implicated in prostate cancer progression and immune suppression. Unlike traditional approaches that encapsulate IL proteins or genes for direct therapeutic effects, this strategy utilizes liposomes to selectively silence the IL30 gene, thereby modulating the tumor microenvironment.

The antibody-conjugated immunoliposomes demonstrated efficient delivery of CRISPR/Cas9 components to prostate cancer cells, achieving high genome-editing specificity and minimal off-target effects. Biweekly administration in preclinical models significantly reduced tumor size, inhibited metastasis, and improved survival. Furthermore, the IL30-targeting liposomes disrupted key oncogenic pathways by downregulating angiogenic and immunosuppressive factors (e.g., VEGFA, CXCL1, and TGFβ1) and upregulating tumor-suppressive genes (e.g., CDH1, DKK3, and PTEN). Importantly, these liposomes also altered the immune landscape of the tumor by suppressing NFKB1 expression and reducing the infiltration of immunosuppressive cells, thereby enhancing antitumor immune responses [201].

This study reinforces the versatility of liposomal delivery systems in cancer therapy by showcasing their ability to mediate targeted gene editing, offering a complementary approach to traditional liposomal interleukin therapies. Integrating this CRISPR-based strategy with existing liposomal interleukin platforms could broaden the scope of immune-based cancer treatments, particularly by targeting interleukin-driven oncogenic pathways.

## 8. Challenges in Liposomal Drug Delivery Systems and Future Perspectives

Over the past two decades, substantial progress has been made in developing anticancer nanoparticles, with many undergoing preclinical and clinical evaluations. However, despite their potential, only a limited number of these technologies have achieved regulatory approval, most of which rely on liposomal platforms [202]. Several obstacles hinder the clinical translation and therapeutic efficacy of liposomal drug delivery systems.

One major challenge is the phenomenon of elevated interstitial fluid pressure in solid tumors, first described by Jain [203]. Elevated IFP presents a significant barrier to efficient drug delivery, as the increased pressure within tumor tissues restricts nanocarrier infiltration. This issue is particularly pronounced at the tumor core, where IFP is highest, resulting in incomplete drug distribution. Various strategies have been proposed to mitigate IFP and enhance drug penetration into tumors, highlighting the importance of addressing this obstacle [204].

Another key limitation arises from the rapid clearance of liposome-associated drugs from the plasma, which diminishes their therapeutic efficacy. Large, charged liposomes are efficiently removed by the liver, while the spleen clears them with a half-life of less than one hour [112,205]. Moreover, the interaction between liposomal components and the immune system further complicates their clinical application. Synthetic modifications designed to improve liposome utility can inadvertently stimulate antibody production against both the liposomal components and the encapsulated drugs. For instance, repeated administration of PEGylated liposomes has been linked to accelerated clearance from the bloodstream, reducing their long-circulating properties [206].

Classical liposomes also face rapid clearance due to the adsorption of plasma proteins (opsonins) onto their exposed phospholipid membrane, which triggers recognition and uptake by the mononuclear phagocytic system (MPS). A significant advancement in liposomal technology was the development of Stealth^®^ liposomes, which incorporate a hydrophilic polymer coating, typically polyethylene glycol (PEG), to evade MPS recognition. This innovation extends the half-life of liposomes from a few minutes to several hours and shifts their pharmacokinetics from dose-dependent and saturable to dose-independent behavior [135].

Further advancements in liposome technology introduced molecular targeting by attaching site-specific ligands to their surface. While this enhances intracellular drug delivery through receptor-mediated endocytosis, it also exposes the internalized material to acidic lysosomal environments and enzymatic hydrolysis. Such degradation reduces the biological activity of sensitive therapeutics, such as nucleic acids and peptides, posing a significant challenge to maintaining drug efficacy [134,207].

In addition to these biological and technical challenges, liposomal formulations face issues during manufacturing, storage, and transport. High-pressure or high-temperature conditions can alter particle crystallinity and damage the formulated drug, while sedimentation, crystal growth, or agglomeration during storage impacts the stability and efficacy of nanoformulated drugs. These challenges necessitate robust manufacturing protocols and advanced quality control measures [208].

The inherent variability of tumors, both among different types and within individual tumors, further complicates the effectiveness of liposome-based drug delivery. This heterogeneity undermines the EPR effect, which is often a cornerstone of nanoparticle targeting strategies. While long-circulating nanoparticles improve pharmacokinetics, their stability may limit cellular drug delivery by reducing drug release and nanoparticle uptake by target cells. These limitations have prompted the exploration of alternative approaches to enhance cellular drug delivery and achieve more precise tumor targeting [209,210,211].

Biological interactions between nanoparticles and the tumor microenvironment (TME) also influence therapeutic outcomes. Factors such as nanoparticle surface characteristics, geometry, size, elasticity, and the presence of targeting ligands affect these interactions and, consequently, the EPR effect. However, much of the current understanding of nanoparticle behavior is derived from animal studies, and its applicability to human systems remains uncertain [212].

The development of liposomal delivery systems is further hindered by high costs and stringent quality control requirements. The complexity of combining multiple components into a single nanoscale carrier increases manufacturing challenges and complicates product reproducibility. Additionally, the targeting efficiency of liposomal nanovesicles remains suboptimal, as similar ligands may exist on various cell types, leading to non-specific biodistribution and toxicity. Efforts to improve liposome targeting and pharmacokinetics are essential for enhancing therapeutic outcomes [213].

Clinical trials for liposomal formulations add another layer of complexity. Formulations must meet rigorous pharmaceutical and commercial quality standards before being evaluated in clinical trials. Even after achieving these milestones, translating the therapeutic efficacy observed in preclinical animal studies to human applications remains a significant hurdle. Moreover, cost–benefit analyses often limit the clinical adoption of certain liposomal therapies, highlighting the need for continued innovation and optimization in this field [51].

By addressing these multifaceted challenges, liposomal drug delivery systems hold the potential to overcome current limitations and achieve broader clinical application.

## 9. Methodology

### 9.1. Search Strategy

To ensure a comprehensive and systematic review, a structured search strategy was employed using the PubMed and Web of Science databases. The following search terms and Boolean operators were applied: (“liposomal interleukin” OR “liposome-encapsulated interleukin”) AND (“gene therapy” OR “gene delivery”) AND (“cancer” OR “tumor” OR “oncology”) AND (“pre-clinical studies” OR “animal models” OR “in vitro studies”).

No language restrictions were applied to capture a global perspective on the subject. The search included studies published from inception until September 2024.

### 9.2. Inclusion and Exclusion Criteria

#### 9.2.1. Inclusion Criteria

Studies focusing on liposomal formulations of interleukins in cancer therapy.Preclinical studies utilizing in vitro or in vivo approaches.Studies addressing the development, optimization, or biological effects of liposomal interleukin delivery systems or gene therapy.Articles present original research, including pharmacokinetics, pharmacodynamics, and therapeutic outcomes.

#### 9.2.2. Exclusion Criteria

Clinical trials and meta-analyses.Studies not related to cancer treatment.Research focuses on non-liposomal formulations of interleukins.

#### 9.2.3. Data Extraction and Analysis

Relevant studies were extracted and organized. Key data fields included the following:Study design and objectives.Type of interleukin and liposomal formulation.Experimental models (e.g., cell lines, animal models).Methods of gene delivery (if applicable).Pharmacological and therapeutic outcomes.Adverse effects and safety profile.

#### 9.2.4. Data Synthesis

A narrative synthesis approach was used to outline the key findings, identify key trends, and discuss advancements in the field. Data were categorized as follows:Overview of liposomal drug delivery systems;Liposomal interleukin therapy;Preclinical studies on liposomal interleukin therapy;Liposomal interleukin gene therapy for cancer treatment.

Acknowledging its potential limitations, this review recognizes the inherent variability in experimental models and the challenges of translating preclinical findings into clinical practice.

## 10. Conclusions

Liposomal IL and liposomal IL gene therapy represent promising advancements in cancer immunotherapy, providing enhanced delivery systems that improve therapeutic effectiveness while reducing systemic toxicity. The use of liposomal carriers, particularly CLs, has shown significant potential in enhancing the pharmacokinetics, stability, and targeting abilities of cytokines such as IL-2, IL-12, and IL-15. These systems not only improve cytokine delivery to tumor sites but also activate robust immune responses, promoting regression of tumors and long-term immunity in preclinical models.

Several challenges persist, particularly in optimizing the EPR effect and addressing tumor heterogeneity, complicating drug delivery. The complexity of liposomal formulations, combined with scalability, manufacturing, and cost issues, stresses the significance of ongoing research and innovation in nanotechnology and gene delivery systems. Despite these obstacles, advancements in liposomal modifications, ligand targeting, and combination therapies with immune modulators are opening new possibilities for enhancing the precision and effectiveness of cancer treatments.

In the future, successful clinical translation will involve navigating regulatory and production challenges, as well as demonstrating evident therapeutic advantages in human trials. The continued improvement of liposomal IL and gene therapy presents great potential to revolutionize cancer immunotherapy, providing a safer and more targeted approach to treating various malignancies.

## Figures and Tables

**Figure 1 pharmaceutics-17-00383-f001:**
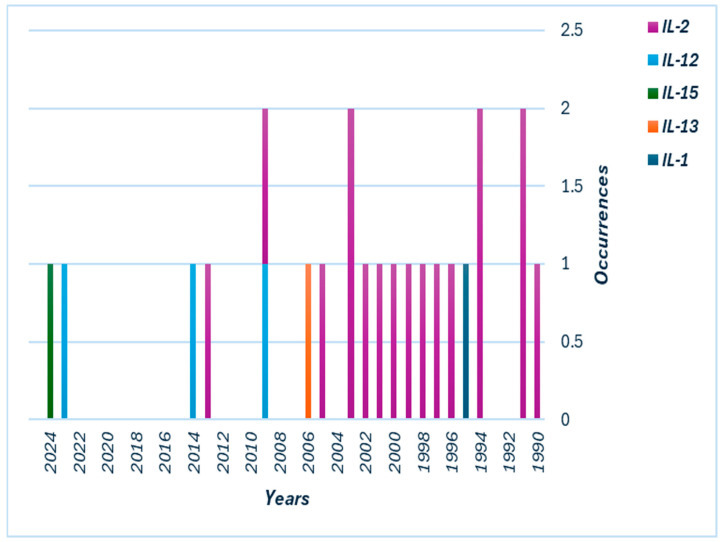
Stacked bar chart showing trends in liposomal interleukin (IL) therapy studies over the years. Each stacked bar represents the occurrence of different ILs (e.g., IL-2, IL-12, IL-1, IL-13, and IL-15) for a given year, indicating how the focus on specific IL therapies has evolved. The color-coded legend identifies the specific IL types included in the studies.

**Figure 2 pharmaceutics-17-00383-f002:**
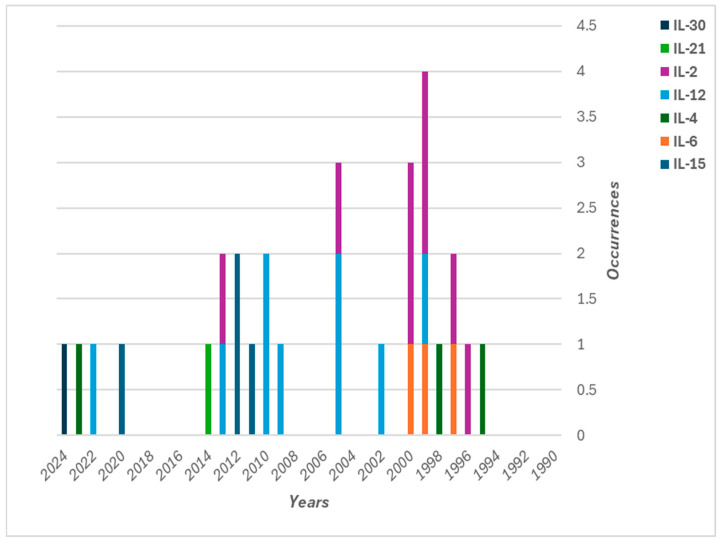
Stacked bar chart shows the distribution of liposomal interleukin gene therapy studies over the years. The chart highlights the use of various IL types (e.g., IL-2, IL-12, IL-15, IL-4, and IL-21). Each stacked bar represents the number of occurrences of specific IL types for a given year, showing how research emphasis on different ILs has shifted over time. The color-coded legend identifies the distinct IL types included in these gene therapy studies.

## Data Availability

No new data were created in this study. Data supporting the reported results are available in the cited references.

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
