# Peer review of "Advances in Liposomal Interleukin and Liposomal Interleukin Gene Therapy for Cancer: A Comprehensive Review of Preclinical Studies"

_pharmaceutics, 2025, doi:10.3390/pharmaceutics17030383_

Round 1

Reviewer 1 Report

Comments and Suggestions for Authors

In this article, the authors reviewed the advances in liposomal interleukin and liposomal Interleukin 2 gene therapy for cancer using PubMed and Web of Science databases. They detailedly introduced the application of interleukins, liposomal drug delivery systems, liposomal interleukin therapy and liposomal interleukin gene therapy for cancer treatments.

Here are some comments and suggestions:

1. Why did you only review preclinical studies of liposomal interleukin and liposomal Interleukin 2 gene therapy for cancer therapy, not including the clinical studies? Sample introduction of clinical trials would give the reader a clearer picture of the potential, progress and current status of liposomal interleukin and liposomal interleukin 2 gene therapy for cancer in clinical.

2. For methodology in section 4, I think it should be put at last. Because it not very important for a review paper and it made the context interrupt.

3. I think the preparation technology of liposomal interleukin and liposomal Interleukin 2 gene is also very important in the preclinical studies. The readers may want to know how they prepared these liposomal interleukin and liposomal Interleukin 2 gene. You should spend several paragraphs to describe the preparation technology/methods of liposomal interleukin and liposomal Interleukin 2 gene.

Reviewer 2 Report

Comments and Suggestions for Authors

In the submitted review manuscript (pharmaceutics-3398199), the authors comprehensively discussed the possibilities of liposomal interleukin and liposomal interleukin gene therapy for treating different types of cancer, based primarily on the results of preclinical studies. They correctly concluded that this type of delivery of immunomodulatory molecules may greatly benefit the future treatment of cancer.

Overall, it is a well-thought-out and implemented review of the available literature on an important and current topic. It uses and cites original scientific studies, which is always a plus and a recommendation in writing review papers.

Since the author's vision and concept must be respected in this type of publication, I have no substantive professional (scientific) objections; they are technical and not necessarily negligible.

1. There is something very wrong with referring to figures and displaying them. What is/where is figure 1, or figure 3, or figure 5? Figures 2, 4, and 6 are shown in the text, but it is unclear what they illustrate, considering that there is no direct reference in the manuscript's text. The authors must check all this in detail and fix/add what is needed!

2. It is unclear why the country where the authors research within their affiliation is not listed (lines 6-11).

3. Although the text is generally nicely integrated into the given format, there were quite a few technical and typing errors (extra or missing spaces, punctuation marks, etc.): lines 66, 75, 112, 133, 172, 245, 326, 403, 415, 426, 488, 497.

4. Latin words like via, in vivo, in vitro are always better written in italics! Please correct (in many places in the text, e.g., lines 207, 218, 251, 313, etc.).

5. Line 253, please specify the month; the entire calendar year is not covered!

6. Please remove unnecessary text on lines 620 and 621 and "personal" messages from the acknowledgments. They have no place there.

Round 2

Reviewer 1 Report

Comments and Suggestions for Authors

The authors made a great effort on the revision of this articles with huge improvement. All my concerns were addressed. I have no other comments to this article and recommend publishing.

Author Response

Comments and Suggestions for Authors

The authors made a great effort on the revision of this articles with huge improvement. All my concerns were addressed. I have no other comments to this article and recommend publishing.

Response: 

Thank you very much for your thoughtful and positive feedback. We greatly appreciate your recognition of the revisions and the improvements made. We are pleased to hear that all of your concerns were addressed. Your recommendation for publication means a lot to us, and we are grateful for your time and effort in reviewing our work.

Reviewer 2 Report

Comments and Suggestions for Authors

The authors made the changes and clarifications requested by the reviewers in the revised manuscript (pharmaceutics-3398199-peer-review-v2). However, the text still needs to be extensively polished technically. For example, it remains unclear why (it will turn out) the Graphic abstract's image is still positioned in the middle of the text or why Figure 2 text was not changed to Figure 1 on line 545. Further, at the request of another reviewer, the newly inserted references are not the most properly cited (e.g., what is in italics). Overall, this is a good Review article; congratulations to the authors.

Round 3

Reviewer 2 Report

Comments and Suggestions for Authors

Everything in this version of the manuscript (pharmaceutics-3398199-peer-review-v4.) is as it should be; congratulations to the authors on a nice publication.